# Medial pivot designs result in improved patient reported outcome measures and range of motion when compared to cruciate retaining total knee replacements: A systematic review and meta-analysis

**Darren Puttock[1], Akhilesh Pradhan[1], Pip Divall[2], Amit Bishnoi[1], Arijit Ghosh[1], Seth O'Neill[3]\*, Randeep Singh Aujla[1]**

**1** Leicester Orthopaedics, University Hospitals of Leicestershire, Leicestershire, United Kingdom, **2** University Hospitals of Leicester, Leicestershire, United Kingdom, **3** School of Healthcare, University of Leicester, Leicestershire, United Kingdom

\* so59@le.ac.uk

## Abstract

### Background

Knee arthroplasty remains one of the most important treatment options in improving quality of life for patients with end-stage knee osteoarthritis. However, roughly 20% of patients remain dissatisfied with their outcome. Perceived implant instability and range of motion are factors that may contribute to dissatisfaction. The medial pivot (MP) total knee replacement (TKR) is postulated to provide increased stability due to greater implant conformity and replication of anatomical function compared to cruciate retaining (CR) implants. This systematic review and meta-analysis evaluates the impact of MP TKR on patient reported outcome measures (PROMs,) range of motion (ROM), pain scores and functional assessment measures in comparison to traditional CR implants.

### Methods and findings

An extensive literature search of multiple databases was conducted to identify eligible high quality studies which compared PROMs data for MP and CR TKR, with a minimum of 12 months follow-up. Our primary outcome was the forgotten joint score (FJS-12). Secondary outcomes included additions PROMs, ROM data and functional assessments. Risk of bias and quality of research were assessed by GRADE rating and the AMQPP tool respectively. A total of 7 articles were included in the systematic review, encompassing 675 patients aged 59–86 years. Four studies assessed FJS-12, with mean difference of 7.46 (−2.44–17.37) in favour of MP TKRs, which was not statistically significant. Overall, 1415 PROM scores from 675 patients were included, giving a statistically significant difference 0.34 (0.16–0.52) and an effect size of 3.69

**Data availability statement:** Contained within the manuscript.

**Funding:** The author(s) received no specific funding for this work.

**Competing interests:** The authors have declared that no competing interests exist.

(p = 0.0002) in favour of MP designs utilising a standardised mean difference analysis. ROM data demonstrated an overall statistically significant mean difference of 4.63° (1.00–8.27) in favour of MP knees. Further functional outcomes, laxity, power measures demonstrated favourable outcomes for MP knees but were ineligible for inclusion in pooled analyses.

## Conclusion

No statistical difference was observed for the majority of PROMs. PROMs including FJS-12, range of motion and functional outcome scores trended towards favouring MP TKRs; with a statistically significance advantage seen for pooled PROMs scores and ROM. However, there remains limited data relating to functional outcome measures within the literature. Further high-powered, multicentre studies are required to analyse whether MP TKRs are superior to CR TKRs regarding functional outcomes.

## Introduction

Total knee replacement (TKR) remains the mainstay of treatment for end stage knee osteoarthritis, offering significant improvement in quality of life and patient reported outcome measures (PROMS) [1,2]. Data from the national joint registry (NJR) demonstrates survivorship of >95% at 10-years, with the majority of UK patients receiving a cruciate retaining (CR) implant design [3]. Despite the ever increasing demand and proven benefits, approximately 20% of patients express dissatisfaction with their post operative outcome [4,5].

There are a multitude of factors which interplay in causing patient dissatisfaction after TKR [6]. Failure to adequately replicate natural knee kinematics is one such contributing factor, and can be influenced by the surgeons choice of implant [7]. Multiple design philosophies have evolved in attempts to address this including single radii, multi-radii, fixed-bearing, mobile-bearing, cruciate-retaining (CR), posterior-stabilised (PS), medially stabilised and medial pivot designs (MP) which attempt to maximise functionality without compromising implant longevity [8,9]. Within the literature, the use of a CR implant has proven benefit in the treatment of end-stage osteoarthritis of the knee, however, the use of MP implants is gaining popularity [10]. Systematic reviews and meta-analyses, published between 2021–2022, suggest that outcomes are comparable between CR, PS and MP designs [10–12]. However a separate review also published in 2022 demonstrated that, MP knee replacements have superior PROMs in comparison to PS TKRs [13].

MP TKRs employ an asymmetrical tibial insert combining a congruent medial compartment with a less conformed lateral compartment, enabling increased femoral rollback and medial stability to produce kinematics which are more reflective of physiological norms [14]. Additionally, this is postulated to result in improved quadricep function, another critical factor influencing patient satisfaction [15].

Despite emerging evidence in favour of MP TKRs regarding PROMs, such as Forgotten Joint Score (FJS), there is limited literature regarding the effect of MP TKRs

upon other functional outcome measures such as stairs assessment, walk tests and timed chair rise, which may provide further information regarding the rehabilitation advantage of MP TKRs [16]. Superiority of a particular implant design would add weight to arguments for their use, in the form of transferable benefits for patient's activities of daily living. In addition, there are no comparison studies within the literature analysing muscle bulk and corresponding effects on lower limb muscle strength in the postoperative period between MP and CR TKRs [17]. The psychological impact of stability is also poorly evidenced in the literature. Whilst MP implants are hypothesised to improve overall stability of the knee joint, their impact of the perception of stability and patient's confidence in their knee has yet to be explored.

This systematic review and meta-analysis compares PROMs and functional outcomes of CR and MP TKRs; in order to ascertain whether there is a rehabilitation and functional advantage in the early post-operative period. We hope to identify deficiencies in the literature whereby future research can be targeted.

## Methods

This systematic review was registered with PROSPERO (CRD42024558554) and conducted in accordance to Preferred Reporting Items for Systematic Review and Meta-Analyses (PRISMA) guidelines.

### Inclusion criteria

- Original research of high methodological quality, as assessed by the Assessing the Methodological Quality of Published Papers (AMQPP) tool, focusing on PROMs and functional outcome measures in CR versus MP TKRs

- Accepted study designs included: randomised (and quasi-) controlled trials, retrospective and prospective cohort studies and case-control studies

- Study participants must be live adults (>18 years old)

- Studies in all languages and from all dates considered

### Exclusion criteria

- Studies related to only PS versus MP knee replacements were excluded

- Studies involving patient populations not undergoing knee arthroplasty were excluded

- Reviews, conference abstracts, proceedings, small case series, case reports, letters, cadaveric research, non-human studies, and laboratory models were also excluded

### Search strategy

An electronic search of the EMBASE, MEDLINE, CINAHL, and Cochrane CENTRAL databases from their inception was conducted. The initial search was performed in July 2024 and repeated in October 2025. Search headings and MeSH terms were formulated, and search terms included 'knee arthroplasty', 'cruciate retaining', 'medial stabilised', 'medial pivot' and 'ball and socket' (detailed search strategy is available in Appendix 1). All article titles were independently screened by two authors (A.P and D.P), and any inconsistencies regarding eligibility were addressed by full text review. Any disagreements were resolved by the senior consultant author panel (A.B and R.A).

### Data collection and analysis

Raw data extracted from studies was recorded on a Microsoft Excel proforma. Data collection included author information, study design, country of origin, participant numbers, gender, anaesthetic protocols, length of stay, PROMs, pain scores

(VAS), functional assessments and range of motion. All quantitative data available for these outcomes was recorded. Where data was missing, two unsuccessful attempts were made to contact corresponding authors to request additional data as required. A.P and D.P conducted the data collection with any discrepancies resolved through discussion with senior author R.A. In situations where authors reported PROMs at multiple timepoints beyond 12 months, only the data representing the longest duration of follow-up for that parameter was used.

### Outcomes

The primary outcome was FJS-12 with secondary outcomes including additional PROMs such as Oxford Knee Score (OKS), Knee Osteoarthritis Score (KOOS) and subtypes and Knee Society Score (KSS). Other secondary outcomes included range of motion, functional assessment measures and pain scores. Range of motion included maximal flexion range and pain scores were mainly reported as VAS scores.

### Data synthesis and statistical analyses

Study, patient demographics, reported outcomes and comparators were systematically summarised. Statistical analysis was conducted using RevMan V5.4 (Cochrane Collaboration, Oxford, United Kingdom). Due to differences in scoring metrics from PROMs data collected, mean standardised difference was calculated using a random effects model with 95% confidence intervals. All disparate outcomes were excluded from the meta-analysis. For ROM, mean difference was calculated using a random effects model with 95% confidence intervals. In both cases heterogenicity was calculated using Cochrane chi-squared test with a standard approach to the interpretation of $I^2$ values obtained.

## Results

Upon conducting a literature search, 603 articles were identified. Following removal of 276 duplicates and rejection of 302 articles based on review of titles and abstracts; 24 articles underwent full text review. As a result, 7 articles were included in the final analysis having met all inclusion criteria and demonstrated high quality research methodology (Fig 1).

### Quality of evidence

The methodological quality was assessed using the AMQPP tool (Table 1). Studies scoring a minimum of 4 out of 6 were deemed eligible for inclusion. Additionally, the quality of the effect estimates in the meta-analysis was evaluated using the GRADE rating (Table 2). Assessments were made by authors AP and DP, with input sought from senior author RSA in cases of disagreement or uncertainty.

Risk of bias in included RCTs was assessed via the Cochrane RoB2 tool. All included trials were assessed as having a low risk of bias.

### Study characteristics

Studies included for analysis were published between 2018–2024 and were conducted in a variety of geographical locations. Two studies were conducted in USA, 2 in Australia, 2 in Japan and 1 study in Denmark. Three studies were prospective randomised controlled trials with a further three being of various retrospective design and one being a prospective cohort study. Further details relating to study characteristics, location and design are available in Table 3. Allowing for the exclusion of posterior stabilised knees from two studies, individual sample sizes ranged from 57–166 patients. In total, 316 cruciate retaining TKRs and 359 medial pivot TKRs were included. The average age of patients included for analysis was 66.89 years, with 62.9% of patients being female.

### Primary outcome measure

Four studies included FJS-12 as an outcome measure, all of which were collected at 12 months post operatively. As a result, 406 FJS scores were included for meta-analysis. Results from three studies were in favour of medial pivot designs,

**PRISMA 2020 flow diagram for new systematic reviews which included searches of databases and registers only**

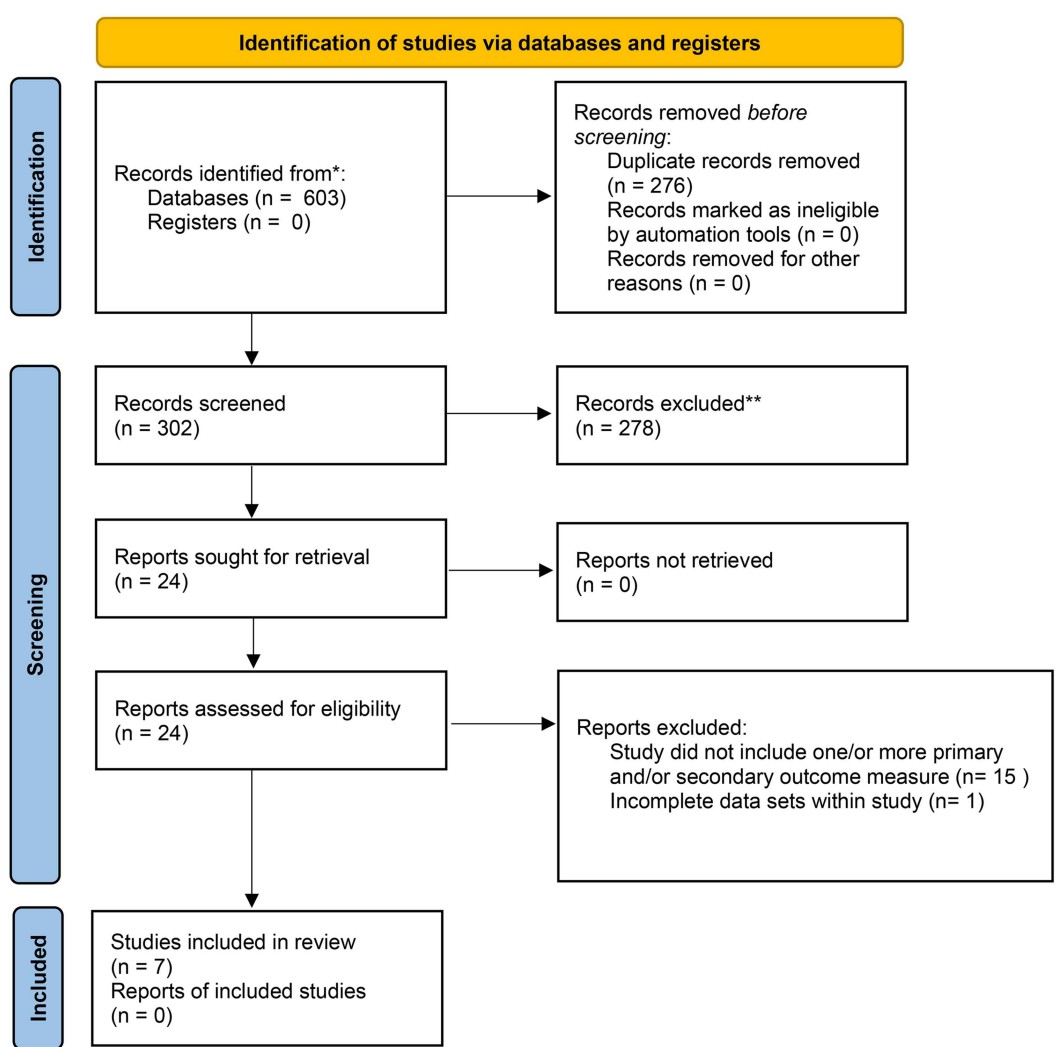

*Consider, if feasible to do so, reporting the number of records identified from each database or register searched (rather than the total number across all databases/registers).

**If automation tools were used, indicate how many records were excluded by a human and how many were excluded by automation tools.

**Fig 1. PRISMA flow diagram for selection of study articles.** From: Page MJ, McKenzie JE, Bossuyt PM, Boutron I, Hoffmann TC, Mulrow CD, et al. The PRISMA 2020 statement: an updated guideline for reporting systematic reviews. BMJ 2021;372:n71. https://doi.org/10.1136/bmj.n71.

two of which were statistically significant. Utilising a mean difference analysis, an overall difference of 7.46 points (−.244–17.37) was found in favour of medial pivot designs, with an overall effect size of 1.48 (p = 0.14). Pooled data results for FJS-12 are demonstrated in Fig 2. Heterogenicity between studies was found to be high ($I^2$ = 73%), with a prediction interval of −22.9–37.87.

**Table 1. Assessment of methodological quality using AMQPP assessment tool.**

| AMQPP Assessment (Yes=1 point, No/not Sure=0 points) | Dubin et al. 2024 [14] | Frye et al. 2021 [15] | Dowsey et al. 2020 [16] | Peterson et al. 2023 [17] | French et al. 2020 [18] | Takahashi et al. 2024 [19] | Nakamura at al. 2018 [20] |
|---|---|---|---|---|---|---|---|
| Is the study original? | Yes | Yes | Yes | Yes | Yes | Yes | Yes |
| Does the study make it clear what it is about? (Hypothesis clearly state, subjects recruited, inclusion and exclusion criteria, circumstances | Yes | Yes | Yes | Yes | Yes | Yes | Yes |
| Is the study design sensible? (what specific intervention was considered and compared? How was the outcome measured?) | Yes | Yes | Yes | Yes | Yes | Yes | Yes |
| Does the study deal with preliminary statistical questions? (sample size, duration of follow up, completeness of follow up) | No | Yes | Yes | Yes | Not clear | No | Yes |
| Does the study avoid or minimise systematic bias? | Yes | No | Yes | Yes | Yes | Yes | Yes |
| Was the assessment blind? (Did the people who assessed the outcome know which group the patient was allocated to?) | No | Not clear | Yes | Yes | Not clear | Not clear | Not clear |
| Total score | 4 | 4 | 6 | 6 | 4 | 4 | 5 |

**Table 2. Assessment of methodological quality using GRADE assessment tool.**

| Outcome | Number of studies (Number of participants) | Risk of bias | Imprecision | Inconsistency | Indirectness | Publication bias | Overall GRADE rating |
|---|---|---|---|---|---|---|---|
| FJS | 4 (386) | Moderate | Low | Low | Low | Low | Very Low |
| OKS | 3 (210) | Moderate | Low | Low | Low | Low | Very Low |
| KSS | 2 (146) | Moderate | Low | Low | Low | Low | Very Low |
| KOOS QOL | 2 (230) | Moderate | Low | Low | Low | Low | Very Low |
| KOOS JR | 2 (213) | Moderate | Low | Low | Low | Low | Very Low |
| KOOS Pain | 2 (230 | Moderate | Low | Low | Low | Low | Very Low |
| Maximum flexion | 2 (176) | Moderate | Moderate | Moderate | Low | Low | Low |
| Total range of motion | 2 (256) | Moderate | High | Moderate | Low | Low | Low |

## Secondary outcome measures

A multitude of additional PROMs were evaluated throughout the different studies including OKS, KSS, KOOS and sub-types of KOOS. Scores utilised on multiple occasions were included for analysis via a mean standardised difference method, resulting in the analysis of 1415 scores drawn from all patients of included studies. As a result, multiple PROMs scores from individual patients will have been included in this analysis. Overall, 1203 scores were collected at the 12-month timepoint, and 213 were collected at 24 months. This resulted in a statistically significant difference of 0.34 (0.16–0.52), with an effect size of 3.69 (p = 0.0002), although with moderate heterogenicity between studies ($I^2$ = 63%). The calculated prediction interval was −0.21–0.74. Considering subgroups of individual scores included results for overall effect were in favour of medial pivot designs for all PROMs, however only results for KOOS -JR reached statistical significance with an effect size of 3.62 (p = 0.0003). Results of this analysis are demonstrated in Fig 3.

Four studies reported data for post-operative ROM, two as maximal flexion and two as overall ROM of the knee joint. The data set included results from 432 patients which was analysed via a mean difference method. Data from 342 patients was collected at 12 months post-operatively and at 24 months from the remaining 90 patients. Overall, a mean

**Table 3. Description of study demographics and characteristics included in analysis.**

| Author (year) | Location | Patient numbers (Male:Female) | Total TKR procedures | Study design | Outcomes measured |
|---|---|---|---|---|---|
| Dubin et al. 2024 | USA | 32:66 | 123 | Retrospective single centre case series | KOOS – JR<br>VAS<br>ROM |
| Dowsey et al. 2020 | Australia | 31:25 | 56 | RCT | OKS<br>KSS<br>WOMAC<br>KSFS<br>Time to get up and go<br>6-minute walk test |
| Takahashi et al. 2023 | Japan | 14:72 | 86 | Retrospective multicentre | FJS<br>AP knee laxity<br>ROM post op |
| Frye et al. 2021 | USA | 73:93 | 166 | Prospective cohort study | FJS<br>KOOS<br>VAS<br>ROM |
| Nakamura et al. 2018 | Japan | 14:76 | 90 | Retrospective multicentre cohort study | KSS<br>ROM |
| French et al. 2019 | Australia | 38:52 | 90 | Single centre RCT | FJS<br>OKS<br>KOOS JR<br>VAS<br>Knee flexion |
| Peterson et al. 2023 | Denmark | 39:25 | 64 | RCT | FJS<br>OKS<br>KOOS<br>Leg extension power<br>MUA for stiffness |

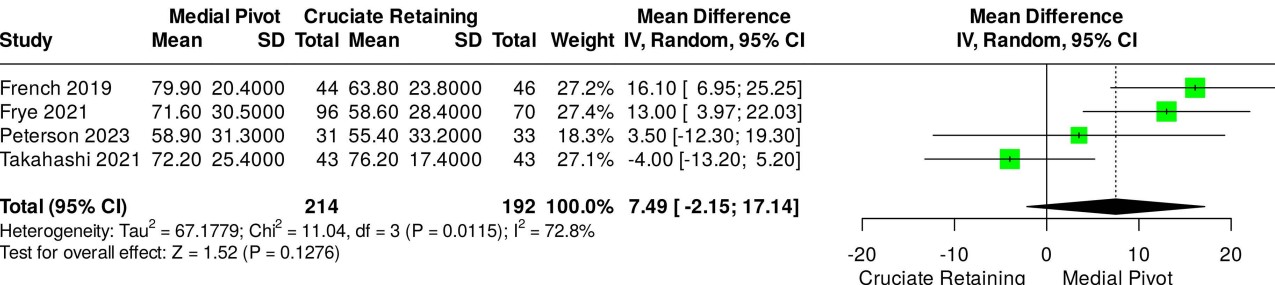

**Fig 2. Meta-analysis and pooled results for FJS-12 comparing CR versus MP knee replacements.**

difference of 4.63° (1.22°-5.82°) was found in favour of MP designs, with an effect size of 2.50 (p = 0.01) and high heterogenicity ($I^2$ = 76%) and a prediction interval of −6.73°–16.0°. This is represented in Fig 4.

## Risk of publication bias

Funnel plots (Fig 5a–5c) indicate no publication bias was present for outcome measures included in meta-analysis, with Eggers test being used to assess for funnel plot asymmetry.

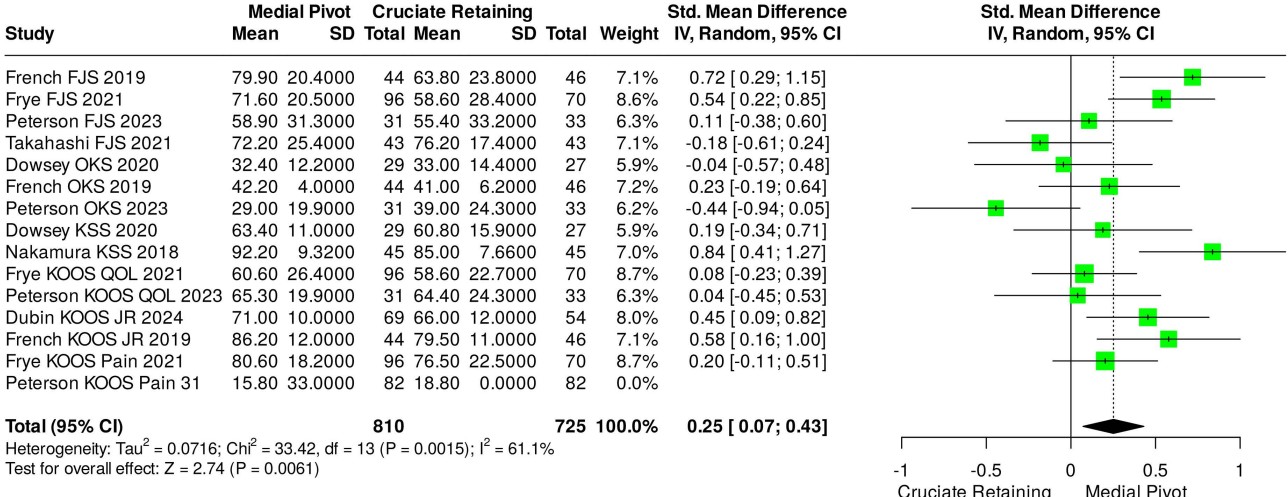

**Fig 3. Meta-analysis and pooled results for all other reported PROMS (excluding FJS-12) comparing CR versus MP knee replacements.**

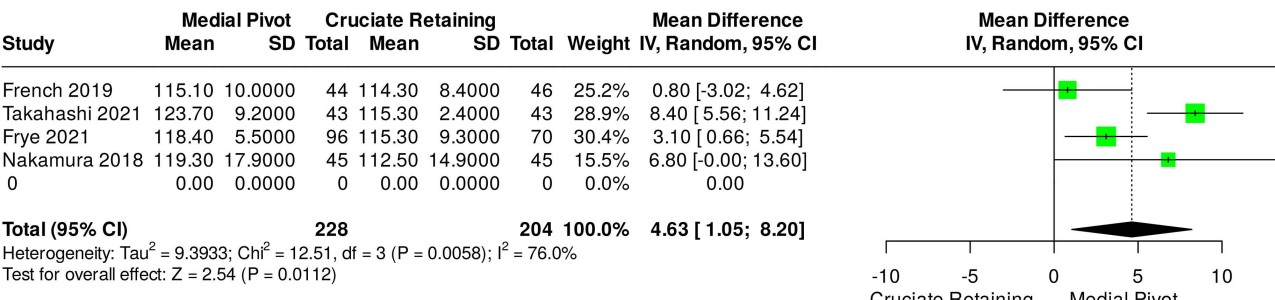

**Fig 4. Pooled data results for studies reporting maximal flexion and overall range of motion after CR versus MP knee replacements.**

Visual analogue scales for pain (VAS) results were reported by three studies, all of which published scores favouring medial pivot designs. Dubin et al. found that by 1-year post procedure VAS scores were significantly lower for patients receiving a MP TKR (1.70 vs 3.76 p < 0.001) [18]. Similarly, Frye et al. observed that VAS scores were lower for MP TKRs at all post operative time points up to 12 months [19].

Additional reported outcomes included functional outcomes such as time to get up and go and 6-minute walk test, post-operative knee extension power and AP knee laxity as well as further PROMs (WOMAC, KSS Satisfaction, KSFS and KOOS ADL). However, each were only reported in singular studies or lacked sufficient data and therefore could not be utilised for pooled analysis.

## Discussion

The findings of this systematic review and meta-analysis demonstrate that during the early postoperative period MP TKRs offer superior outcomes in terms of PROMs and joint ROM, compared to widely used CR TKRs. Similar results have been obtained by single centre studies of both cohort and quasi-randomised design, reporting on selected PROMs [18,19,21]. Pooled data from included studies found significant differences in KOOS JR scores in favour of MP TKRs, with outcomes for combined FJS and KSS scores nearing, but not reaching significance.

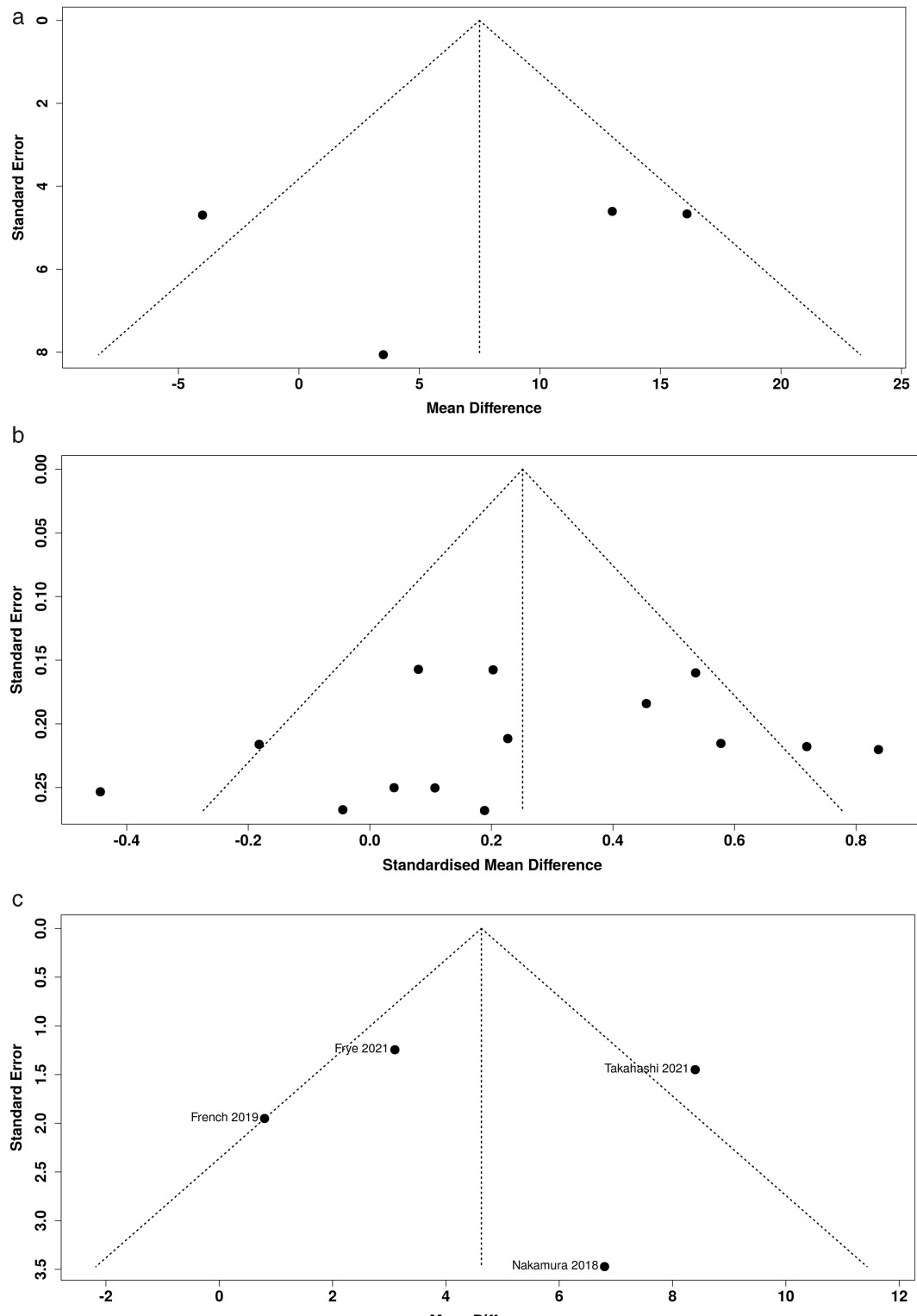

**Fig 5. Funnel plots comparing outcomes across studies. a**: Funnel plot for Forgotten Joint Score data, **b**: Funnel plot for pooled PROMs data, **c**: Funnel plot for Range of Motion data.

Four studies reported FJS-12 for their cohorts, three of which higher mean scores for MP TKRs [19,21,22]. The results obtained by French et al. and Frye et al. individually reached significance [19,21] whilst Takahashi et al. found no significant differences between cohorts [23]. Meta analysis revealed a mean difference of 7.46 points (95% confidence interval −2.44–17.37). This does however fall below recently established values for the minimally important clinical difference (MCID) for the FJS, although this was not established specifically for comparison between cohorts receiving different TKR implant types [24].

Despite being unable to demonstrate statistical or clinical significance, there was a clear trend in data favouring MP TKRs when observing FJS-12. No included study favoured CR TKRs for this outcome and further studies wound benefit in observing this PROM to further aid pooled meta-analysis.

No included study analysed the FJS (or other PROMS) in the context of substantial clinical benefit (SCB), which has been defined for the FJS in the context of TKA as a change of 28 points [20]. SCB is increasingly recognised as an important parameter as it considers the change in symptom state from the patient's perspective and determines whether they can perceive a considerable improvement following an intervention. Many advocate for its use over alternative methods such as MCID, as this is felt to reflect a 'floor value' rather than being a measure of success. Analysis, in the context of whether a particular implant results in SCB may be more beneficial and produce results translatable to patients and reflective of achieving success, thus better informing clinical practice [25,26]. However, in the context of comparison of two implant types with proven success and approval from centralised governance bodies such as the Orthopaedic Devices Evaluation Panel (ODEP), it is possible that both will demonstrate SCB.

Analysis of all utilised PROMs, via a mean standardised difference approach did reveal a significant difference in favour of MP TKRs in comparison to CR. Overall, 10 out of 15 mean PROMs scores obtained by included papers favoured MP TKRs when collected at or beyond 12-months of follow up.

Dubin et al. provide comprehensive follow up data using the KOOS JR score, which demonstrates a statistically significant superiority of the MP TKR over the CR TKR previously used in this centre, which was consistent at multiple post-operative timepoints from 2 months, up to 2 years. These results provide information from a PROM designed specifically for the population of interest [27]. The same author also demonstrated a significantly improved range of motion over the same period in favour of MP TKRs. Although such results are encouraging and support the increasing use of MP TKRs, caution should be exercised as they are drawn from a single surgeon series which has been retrospectively analysed [18]. Furthermore, due to lack of complete data, the results pertaining to ROM unfortunately could not be utilised for meta-analysis. Similarly, Takahash et al. published results obtained from a multicentre study, which also demonstrate significantly superior post-operative knee flexion in the MP group, concurring with findings from Dubin and others [18,23,28].

Pooled data included in this study indicates MP TKRs afford patients an increased ROM, which is statistically significant beyond one year post procedure. To the best of our knowledge, the minimal clinically important difference for ROM in this patient population has yet to be defined. Systematic review data exists, quantifying the minimum clinically important change for conservatively treated knee osteoarthritis, which defines the MCIC as being between 3.8–6.4° [29]. The mean difference found in this review was 4.8° in favour of MP TKRs, therefore it could be argued this represents a meaningful advantage, however the heterogenicity in follow up duration, measurement methodology and interventions in included studies compared with those used to define this MCIC is noted. Further work to better define the MCIC or MCID for a post-operative cohort would be of benefit for future research and may also allow an increased understanding of findings already available within the literature. It would not however, be unreasonable to suggest that a greater ROM results in both a perceived and actual functional advantage for patients and reflect in their psychological perception of the implant.

Two previous systematic reviews are available comparing MP with CR TKRs. One such review yielded larger patient numbers as authors included studies also comparing MP TKRs with PS, and mobile bearing implants in contrast to our own methodology [11]. In doing so they found comparable outcomes in terms of overall PROMs data. However their subgroup analyses findings for MP vs CR TKRs using FJS-12 and KSS scores align with our own findings, although such

findings are drawn from only two retrospective studies which both also met the inclusion criteria for this review [21,28]. An earlier published review, which only included two studies in their meta-analyses found no clear consensus, with standardised mean difference analysis for KSS scores favouring MP TKRs, however when conducted on WOMAC scores, non-MP TKRs demonstrated superiority [30].

Functional measures were rarely utilised by studies included in this review and those which were used were only employed by single studies. As such, conducting any form of pooled analysis was not feasible. Measures such as the six minute walk test (6MWT) and time to get up and go can be argued to represent real life, day to day function post TKR, and have been validated for use as well as having MCID defined [16,31]. Dowsey et al. reported on both these parameters, however no statistically significant differences in improvement from baseline was observed at either 6 or 12 months post procedure [32]. Differences in baseline characteristics were reported by authors, including a higher BMI and higher proportion of multiple co-morbidities in the MP group, which could explain both the lower pre-operative distance covered in the 6MWT test by this group and the lack overall difference found. Such variations in baseline characteristics may have confounded the results obtained. Further studies including these measures would be of use to clarify these results and potentially identify any functional advantage which may exist dependent on implant design. Previously, use of the 6MWT has been recommended within the first 4 months post TKA, therefore the timings of assessments were not likely to detect any advantage that may have been present [33].

Peterson et al. assessed post-operative knee extension power, demonstrating no difference at 12-month post procedure. This could also be considered an important variable to explore given that it has previously been correlated with functional outcomes following TKA [22,34]. The lack of difference observed may be due to the timing of data collection, with previous studies suggesting that deficiencies in extensor muscle strength post TKA persist for only around three months [35].

No included study specifically analysed the psychological impact of either TKR design. Dowsey et al. included veterans RAND 12-item health survey and found that for its mental component there was a statistically significant advantage in favour of MP TKRs. This provides limited data to suggest such implants could confer a psychological advantage over alternative designs [32]. However, our authorship group specifically wished to seek reports of patient confidence and perceived stability with their knee replacements, which has not reported in the existing literature. It is well acknowledged that dissatisfaction with knee replacements has significant biopsychosocial components, contributing to overall patient experience [36]. Therefore, it is pertinent that future studies analyse the overall impact of TKR on patient's mental health and their perceived postoperative confidence and stability, aiming to detect possible advantages of one implant type over another within these domains.

Limitations of this systematic review and meta-analyses are noted. The variation in PROMs used by included papers limited analysis to a standardised mean difference approach. Additionally, we note that our methodology for analysis of pooled PROMs has resulted in multiple scores from individual patients being included which may have biased the outcome obtained. Furthermore, the moderate to high level of heterogenicity is noted. Although all PROMs used have their merits, and have been validated for use, a more standardised approach would be of benefit as the evidence base around these implants continues to grow. Our literature search and inclusion criteria enabled us to increase the number of studies included comparing MP and CR designs specifically, in comparison to previously analyses, however it is acknowledged that there remains a lack of representation from European and UK based studies. Future studies from these geographical locations would be on interest to determine the population specific results of such implants. However, to definitively state superiority of one design type over another remains a challenge in the context of a lack of defined and accepted core outcome set of this patient group [37]. Finally, the majority of data available relates to 12- or 24-months post procedure. Ongoing observation and analysis is imperative to monitor the longer-term results and survivorship of these implant designs.

## Conclusion

This systematic review and meta-analysis provides evidence that during the early post-operative period, MP TKRs offer superior outcomes when compared to commonly used CR designs, both in terms of PROMs and range of motion. Sparse data exists relating to functional outcome measures, longer-term results and biopsychosocial factors. As such this should be an area of focus for future work. Further studies will add to the evidence base and should aim to include populations not previously represented to identify any potential variations in outcomes.

## Acknowledgments

Mr Han Hong Chong for statistical analysis guidance.

## Author contributions

**Conceptualization:** Darren Puttock, Akhilesh Pradhan, Pip Divall, Arijit Ghosh, Amit Bishnoi, Randeep Singh Aujla.

**Data curation:** Darren Puttock, Akhilesh Pradhan, Pip Divall, Amit Bishnoi, Seth O'Neill, Randeep Singh Aujla.

**Formal analysis:** Darren Puttock, Akhilesh Pradhan, Pip Divall, Arijit Ghosh, Randeep Singh Aujla.

**Investigation:** Pip Divall, Arijit Ghosh, Amit Bishnoi, Randeep Singh Aujla.

**Methodology:** Darren Puttock, Akhilesh Pradhan, Pip Divall, Arijit Ghosh, Amit Bishnoi, Seth O'Neill, Randeep Singh Aujla.

**Project administration:** Darren Puttock, Akhilesh Pradhan, Pip Divall, Arijit Ghosh, Seth O'Neill, Randeep Singh Aujla.

**Resources:** Darren Puttock, Akhilesh Pradhan, Pip Divall, Arijit Ghosh, Amit Bishnoi, Seth O'Neill, Randeep Singh Aujla.

**Software:** Darren Puttock, Akhilesh Pradhan, Pip Divall, Arijit Ghosh, Amit Bishnoi, Seth O'Neill, Randeep Singh Aujla.

**Supervision:** Seth O'Neill, Randeep Singh Aujla.

**Validation:** Darren Puttock, Akhilesh Pradhan, Pip Divall, Arijit Ghosh, Seth O'Neill, Randeep Singh Aujla.

**Visualization:** Akhilesh Pradhan, Amit Bishnoi, Seth O'Neill, Randeep Singh Aujla.

**Writing – original draft:** Darren Puttock, Akhilesh Pradhan, Amit Bishnoi, Randeep Singh Aujla.

**Writing – review & editing:** Darren Puttock, Akhilesh Pradhan, Pip Divall, Amit Bishnoi, Seth O'Neill, Randeep Singh Aujla.

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
