## [Decision Letter · Decision Letter 0]

25 Sep 2025

Dear Dr. O'Neill,

We look forward to receiving your revised manuscript.

Kind regards,

Sarah Jose, Ph.D.

Staff Editor

PLOS ONE

Journal Requirements:

2. We notice that Figure 1 is uploaded with the file type “Other”, to which you refer in your text on pages 10 and 11. Please amend the file type to “Figure”. If the figure is no longer to be included as part of the submission please remove all reference to it within the text.

Reviewers' comments:

Reviewer's Responses to Questions

**Comments to the Author**

1. Is the manuscript technically sound, and do the data support the conclusions?

Reviewer #1: Partly

Reviewer #2: Yes

2. Has the statistical analysis been performed appropriately and rigorously?

Reviewer #1: Yes

Reviewer #2: Yes

3. Have the authors made all data underlying the findings in their manuscript fully available?

Reviewer #1: Yes

Reviewer #2: Yes

4. Is the manuscript presented in an intelligible fashion and written in standard English?

Reviewer #1: Yes

Reviewer #2: Yes

Reviewer #1: I thank the Editor for the invitation to review this manuscript. I was co-first author on the included Dowsey 2020 article and I wish to make it clear that my comments are in no way influenced by this - I make no specific comments regarding that article in terms of giving it any sort of special treatment.

I believe this is a mostly methodologically sound review and a good fit for the journal, however I have a number of comments I believe need to be addressed prior to recommending acceptance for publication.

- Could be made clearer in the Introduction re: when the most recent systematic reviews were published, to help justify this new review

- AMPQQ acronym (and all acronyms, for that matter) needs to be defined at first use

- Inclusion criteria included studies of high quality. Therefore, studies underwent critical appraisal then low quality studies were excluded? I do not recommend this approach. Essentially this constitutes a sensitivity analysis from the outset

- - Related to this: critical appraisal should be in Results, not Methods. I understand why it is currently in the Methods section, but I disagree this methodological choice (i.e. using critical appraisal as part of the article screening process). A systematic review should include all relevant literature, and then the low quality studies can potentially be excluded in a sensitivity analysis but would still be included in the review overall.

- Was forward and backward citation chasing conducted?

- Has the search been re-run since July 2024? It has now been 12 months so this would be recommended. However, citation chasing may be a potentially acceptable alternative if conducted now to identify more recent publiations.

- I recommend reporting prediction intervals and emphasising these over i-squared values when reporting on heterogeneity. Relevant references: Borenstein 2023 - Avoiding common mistakes in meta-analysis: Understanding the distinct roles of Q, I-squared,tau-squared, and the prediction interval in reporting heterogeneity; IntHout 2016 - Plea for routinely presenting prediction intervals in meta-analysis

- Please provide more detail on the GRADE synthesis methodology, especially considering GRADE should be used to synthesise the entire body of evidence rather than each individual paper (which is what table 2 suggests)

- Suggest using separate risk of bias tool for RCTs

- Was there any plan for narrative synthesis, or only meta-analysis?

Reviewer #2: Dear Authors,

Thank you for submitting this interesting systematic review and meta-analysis.My suggestions to improve the article are as follows:

1. Discuss substantial clinical benefit, which was initially proposed by Glassman et al. for the PROMS as the absence of SCB in the selected studies is a limitation.The CORR study by Lyman et al : What Are the Minimal and Substantial Improvements in the HOOS and KOOS and JR Versions After Total Joint Replacement? Clin Orthop Relat Res. 2018 Dec;476(12):2432-2441 mentions (MCID) using distribution- and anchor-based approaches and the difference that can be considered a large improvement in joint health (substantial clinical benefit) using an anchor-based approach.Surprisingly this article is also missing from the references and the findings of this article are very relevant to this review.

2. A funnel plot is highly desirable to check for publication bias.

Reviewer.

**Do you want your identity to be public for this peer review?** For information about this choice, including consent withdrawal, please see our Privacy Policy

Reviewer #1: **Yes:** Daniel Gould

Reviewer #2: **Yes:** Prof. Roop Bhushan Kalia

---

## [Author Response · Author response to Decision Letter 1]

1 Dec 2025

Response to Reviewers

Dear reviewers,

We sincerely thank you for your time and input reviewing our manuscript. We have considered all points raised and incorporated your valuable input into the revised version. Please see below our responses to each point raised.

Reviewer 1:

1. Reviewer: Could be made clearer in the Introduction re: when the most recent systematic reviews were published, to help justify this new review

Response: We thank you for this suggestion and have clarified the publications years of the quoted evidence accordingly.

2. Reviewer: AMPQQ acronym (and all acronyms, for that matter) needs to be defined at first use

Response: Noted, thank you for highlighting this, AMQPP now defined within text.

3. Reviewer: Inclusion criteria included studies of high quality. Therefore, studies underwent critical appraisal then low quality studies were excluded? I do not recommend this approach. Essentially this constitutes a sensitivity analysis from the outset

Response: Whilst we appreciate the concerns regarding the inclusion criteria, this approach was agreed with all authors within the group in order to draw conclusions only from robust, unbiased trials. Such an approach is common within this field and as such we feel this is acceptable. Although this does potentially constitute a sensitivity analysis, no study ended up being excluded on the basis of quality alone.

4. Reviewer: Related to this: critical appraisal should be in Results, not Methods. I understand why it is currently in the Methods section, but I disagree this methodological choice (i.e. using critical appraisal as part of the article screening process). A systematic review should include all relevant literature, and then the low quality studies can potentially be excluded in a sensitivity analysis but would still be included in the review overall.

Response: Many thanks for highlighting this concern. Text and tables relative to critical appraisal of identified studies has been moved to the results to avoid confusion relating to our methodology.

5. Reviewer: Was forward and backward citation chasing conducted?

Response: Yes, forward and backwards citation chasing was conducted.

6. Reviewer: Has the search been re-run since July 2024? It has now been 12 months so this would be recommended.

Response: The search was repeated on 22/10/2025 and summarised in the PRISMA diagram. 25 new potentially eligible articles were retrieved, with 24 excluded on abstract screening. One article underwent full text screening however was excluded on the basis of data reporting methodology.

7. Reviewer: I recommend reporting prediction intervals and emphasising these over i-squared values when reporting on heterogeneity. Relevant references: Borenstein 2023 - Avoiding common mistakes in meta-analysis: Understanding the distinct roles of Q, I-squared,tau-squared, and the prediction interval in reporting heterogeneity; IntHout 2016 - Plea for routinely presenting prediction intervals in meta-analysis

Response: Thank you for highlighting this to us, we have since calculated the prediction intervals and included these within the results section. Due to the fact that I2 is commonly used we have opted to also retain these within the manuscript but appreciate the merits of prediction intervals over I2 values, hence their inclusion.

8. Reviewer: Please provide more detail on the GRADE synthesis methodology, especially considering GRADE should be used to synthesise the entire body of evidence rather than each individual paper (which is what table 2 suggests)

Response: The Grade synthesis used was determine the certainty of results obtained for each patient reported outcome measure utilised in the analysis. Decisions made regarding each domain were reached via discussion and analysis of each study by authors AP and DP. Where there was disagreement or uncertainty, an opinion from senior author RSA was sought. Table 2 demonstrates this process being conducted for each outcome measure individually and include information relating to the number of studies which included this measure as well as the number of patients for whom a result was obtained. Further information on this process added to manuscript as shown by tracked changes.

9. Reviewer: Suggest using separate risk of bias tool for RCTs

Response: Thank you for this suggestion, risk of bias for included RCTs has been conducted using ROB2 tool and outlined within methods section.

10. Reviewer: Was there any plan for narrative synthesis, or only meta-analysis?

Response: From the outset our plan was purely for meta-analysis.

Reviewer 2:

1. Reviewer: Discuss substantial clinical benefit, which was initially proposed by Glassman et al. for the PROMS as the absence of SCB in the selected studies is a limitation.The CORR study by Lyman et al : What Are the Minimal and Substantial Improvements in the HOOS and KOOS and JR Versions After Total Joint Replacement? Clin Orthop Relat Res. 2018 Dec;476(12):2432-2441 mentions (MCID) using distribution- and anchor-based approaches and the difference that can be considered a large improvement in joint health (substantial clinical benefit) using an anchor-based approach.Surprisingly this article is also missing from the references and the findings of this article are very relevant to this review.

Response: We would like to thank you for drawing this analysis method to our attention and agree its use and context is of value to this review and research area as a whole. Therefore a passage discussing substantial clinical benefit, its merits and potential use within the field has been added.

2. Reviewer: A funnel plot is highly desirable to check for publication bias.

Response: Thank you for this suggestion, funnel plots have been added to the results section of the manuscript.

---

## [Decision Letter · Decision Letter 1]

14 Jan 2026

MEDIAL PIVOT DESIGNS RESULT IN IMPROVED PATIENT REPORTED OUTCOME MEASURES AND RANGE OF MOTION WHEN COMPARED TO CRUCIATE RETAINING TOTAL KNEE REPLACEMENTS: A SYSTEMATIC REVIEW AND META-ANALYSIS

PONE-D-25-32016R1

Dear Dr. O'Neill,

We’re pleased to inform you that your manuscript has been judged scientifically suitable for publication and will be formally accepted for publication once it meets all outstanding technical requirements.

Kind regards,

Dimitrios Sokratis Komaris, Ph.D

Academic Editor

PLOS One

Additional Editor Comments (optional):

Reviewers' comments:

Reviewer's Responses to Questions

**Comments to the Author**

Reviewer #1: All comments have been addressed

Reviewer #3: All comments have been addressed

2. Is the manuscript technically sound, and do the data support the conclusions?

Reviewer #1: Yes

Reviewer #3: Yes

3. Has the statistical analysis been performed appropriately and rigorously?

Reviewer #1: Yes

Reviewer #3: Yes

4. Have the authors made all data underlying the findings in their manuscript fully available?

Reviewer #1: Yes

Reviewer #3: Yes

5. Is the manuscript presented in an intelligible fashion and written in standard English?

Reviewer #1: Yes

Reviewer #3: Yes

Reviewer #1: I thank the authors for their comprehenasive responses to reviewers' comments, and the accompanying thorough revisions to the manuscript.

There are a some aspects of the approach to this review which I personally would have treated differently. However, the authors have provided sufficient responses to my queries and adequately justified their methodological approach. I feel the result is a manuscript that is now suitable for publication, despite not necessarily being the same as I would have written. The authors have provided an extensive discussion and drawn appropriate conclusions.

The omission of a funnel plot from the original submission was, in my opinion, justified considering there were fewer than 10 studies included in the meta-analysis. However, I commend the authors on the inclusion of a funnel plot at the request of the other reviewer.

I look forward to reading this review in its published form.

Reviewer #3: The authors have adequately addressed the reviewers' questions from the first round. I have no further comments.

**Do you want your identity to be public for this peer review?** For information about this choice, including consent withdrawal, please see our Privacy Policy

Reviewer #1: No

Reviewer #3: No

---

## [Editor Report · Acceptance letter]

PONE-D-25-32016R1

PLOS One

Dear Dr. O'Neill,

I'm pleased to inform you that your manuscript has been deemed suitable for publication in PLOS One. Congratulations! Your manuscript is now being handed over to our production team.

Kind regards,

on behalf of

Dr. Dimitrios Sokratis Komaris

Academic Editor

PLOS One